# Using a Fibrinolysis Delivery Catheter in Pulmonary Embolism Treatment for Measurement of Pulmonary Artery Hemodynamics

**Abdelrahman Elhakim** [1,*] **, Martin Knauth** [1] **, Mohamed Elhakim** [2] **, Ulrich Böhmer** [1] **, Johannes Patzelt** [1] **and Peter Radke** [1]

[1] Cardiology Department, Schoen Hospital, 23730 Neustadt in Holstein, Germany
[2] Critical Care and Anesthesia, Nepean Hospital, Kingswood, Sydney, NSW 2747, Australia
[*] Correspondence: aelhakim@schoen-klinik.de or ayelhakim1985@yahoo.com; Tel.: +49-1638542698

**Highlights:**

**What are the main findings?**

- This study demonstrated that ultrasound-facilitated catheter-directed fibrinolysis for the treatment of 45 patients with intermediate high-risk pulmonary embolism was associated with significant improvements in hemodynamics and low major bleeding events as demonstrated in previous trials (Ultima and Seattle II). However, this protocol used a lower dose and shorter duration of t-PA. The third arm of Optalyse PE-Trial used the same regime (6 mg/6 h/lung) for the treatment of 24 patients.
- It focuses on pulmonary artery hemodynamics as a primary precise surrogate marker for therapy effectiveness in addition to RV/LV ratio that was used in previous trials (Ultima and Seattle II), which has a bias of interobserver variability.

**What is the implication of the main finding?**

- It assessed pulmonary artery hemodynamics for the first time by fibrinolysis delivery catheter, without additional need for a right heart catheterization. The results matched the right heart catheter results in EKOS and Heparin arm of Ultima trial, thereby confirming the validity of these diagnostic tools.
- It's a valid and precise diagnostic tool to assess therapy effectiveness and reduce additional procedure-related complications, hospital residency, and economics. It gives this therapy additional diagnostic potential. It stressed the importance of interdisciplinary teams in the management of PE and evaluated the quality of life of these patients. This protocol shortens ICU stay to 6 h.

**Abstract:** Background: Ultrasound-facilitated and catheter-directed low-dose fibrinolysis (EKOS) has shown favorable hemodynamic and safety outcomes in intermediate- to high-risk pulmonary embolism (PE) cases. Objectives: This prospective single-arm monocentric study assessed the effects of using a delivery catheter for fibrinolysis as a novel approach for acute intermediate- to high-risk patients on pulmonary artery hemodynamics PE. Methods: Forty-five patients (41 intermediate–high and 4 high risk) with computer tomography (CT)-confirmed PE underwent EKOS therapy. By protocol, a total of 6 mg of tissue-plasminogen activator (t-PA) was administered over 6 h in the pulmonary artery (unilateral 6 mg or bilateral 12 mg). Unfractionated heparin was provided periprocedurally. The primary safety outcome was death, as well as major and minor bleeding within 48 of procedure initiation and at 90 days. The primary effectiveness outcomes were: 1. to assess the difference in pulmonary artery pressure from baseline to 6 h post-treatment as a primary precise surrogate marker, and 2. to determine the echocardiographic RV/LV ratio from baseline to 48 h and at 90 days post-delivery. Results: Pulmonary artery pressure decreased by 15/6/10 mmHg ($p < 0.001$). The mean RV/LV ratio decreased from $1.2 \pm 0.85$ at baseline to $0.85 \pm 0.12$ at 48 and to $0.76 \pm 0.13$ at 90 days ($p < 0.001$). Five patients (11%) died within 90 days of therapy. Conclusions and Highlights: Pulmonary artery hemodynamics were assessed using a delivery catheter for fibrinolysis, which is reproducible for identifying PE at risk of adverse outcomes. The results matched the right heart

catheter results in EKOS and Heparin arm of Ultima trial, thereby confirming the validity of this potential diagnostic tool to assess therapy effectiveness and thereby reduce additional procedure-related complications, hospital residency, and economics. These results stress the importance of having an interdisciplinary team involved in the management of PE to evaluate the quality of life of these patients and this protocol shortens ICU admission to 6 h.

**Keywords:** intermediate- to high-risk thrombotic pulmonary embolism; ultrasound-facilitated; catheter-directed; low-dose fibrinolysis; pulmonary embolism hemodynamics; pulmonary artery pressure; cohort study

---

## 1. Introduction

Pulmonary embolism (PE) is the third most common cardiovascular cause of death, after myocardial infarction and stroke. The annual incidence is increasing (in Germany from 85/100,000 cases in 2005 to 109/100,000 cases in 2015) [1]. Computer tomography pulmonary angiography (CTPA) is the gold standard for diagnosing PE. However, it has diagnostic limitations, as several pulmonary vessel diseases (congenital pulmonary artery branch stenosis, large-vessel pulmonary artery vasculitis, chronic thromboembolic pulmonary hypertension (CTEPH), and pulmonary artery sarcoma (PAS)) mimic PE and are characterized by multiple filling defects in pulmonary circulation. Therefore, supplemental advanced imaging (Magnetic resonance angiography (MRA), positron emission tomography (PET), and endobronchial ultrasound (EBUS)) and even biopsy may be required to establish a diagnosis [2].

Risk stratification to identify intermediate- and high-risk PE is of paramount significance due to the increased mortality of up to 30% in these groups, according to the European Society of Cardiology—Guideline 2019 [3]. An aggressive interventional therapy administered as early as possible rather than anticoagulation alone might improve the hemodynamic and clinical outcomes (Figure 1).

The advanced treatment options used for PE vary by institution, medical specialty, and operator experience [4]. Additionally, variations or ambiguity in treatment recommendations in clinical guidelines published by societies such as the ESC [3], American Heart Association (AHA) [5], and the American College of Chest Physicians (ACCP) [4,6] and/or the lack of robust clinical trials make advanced treatment decisions challenging.

Treatment modalities of thrombotic pulmonary embolism include anticoagulation, intravenous or catheter-directed thrombolytic therapy, catheter-directed mechanical thrombectomy, and surgical embolectomy.

Due to its favorable hemodynamic results in comparison to anticoagulation alone and superior safety outcomes compared to systemic fibrinolysis, the use of ultrasound-facilitated and catheter-directed low-dose fibrinolysis (EkoSonic Endovascular System, Boston Scientific, Marlborough (EKOS)) could be a reasonable treatment modality in intermediate- and high-risk pulmonary embolism. The results of improvement in hemodynamic effects and favorable safety have been shown in Ultima study [7]. EKOS therapy is FDA-approved for the treatment of acute PE [6].

In Vitro investigation ultrasound separates fibrin strands, enhancing the penetration of the thrombolytic agent into the emboli [8]. In this study, low-dose fibrinolysis was administered locally with substantially lower doses to reduce bleeding complications while at the same time achieving a measurable early reduction in pulmonary artery pressure. These effects might result in a reduction in future events (i.e., recurrent PE).

This article discusses a prospective single-arm monocentric cohort study conducted to evaluate the hemodynamic effect of EKOS using a delivery catheter for fibrinolysis for the first time as a new approach for measuring pulmonary artery pressure for acute pulmonary embolism. This can improve our understanding of invasive and noninvasive hemodynamic measurements as well as outline the safety outcomes of EKOS therapy and the possibility to

combine these data with other interventional modalities, thus influencing current practice and guidelines.

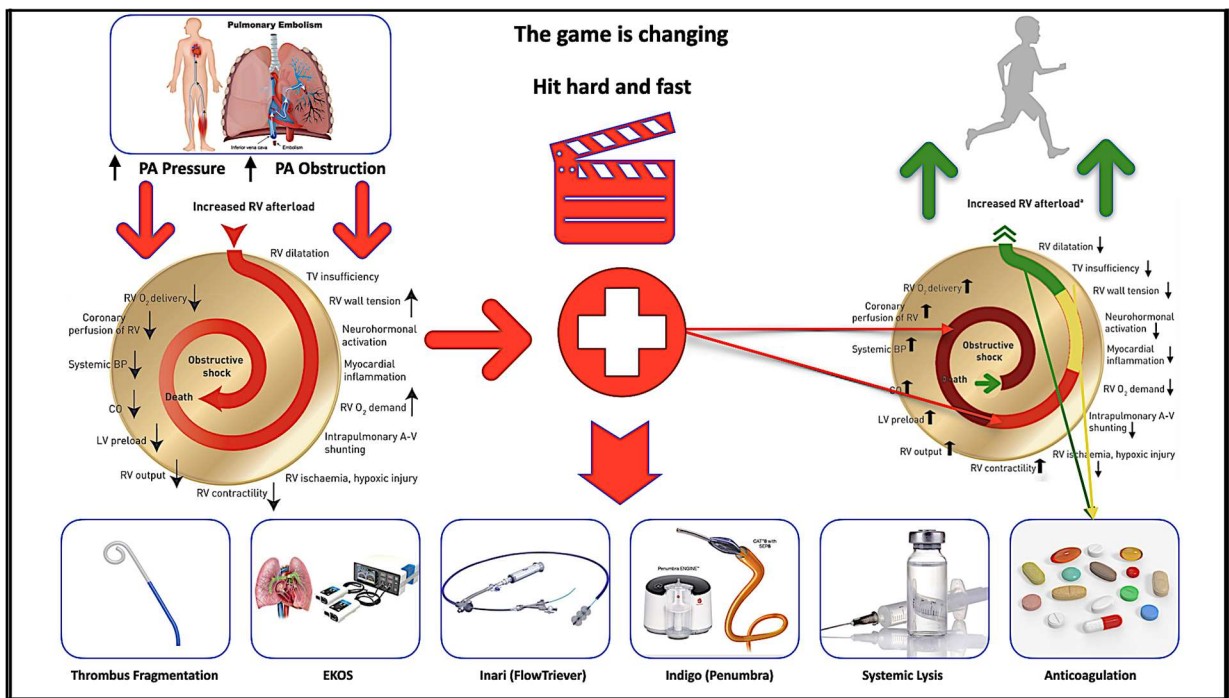

**Figure 1.** Central illustration demonstrating pulmonary embolism process. Shown is the mechanism of hemodynamic collapse and death in acute pulmonary embolism cases and how each risk group is treated (Braun = high risk; red = intermediate–high risk; yellow = intermediate–low risk; green = low risk) (modified from Konstantinidis et al. and ESC Guideline 2019 with permission). PA = Pulmonary artery; A-V = arterio-venous; BP = blood pressure; CO = cardiac output; LV = left ventricular; O2 = oxygen; RV = right ventricular; TV = tricuspid valve; Braun = high risk; red = intermediate–high risk; yellow = intermediate–low risk; green = low risk of PE.

## 2. Methods

### 2.1. Study Design

From August 2020 to April 2022, we screened 63 hospitalized patients with acute high-risk and intermediate- to high-risk PE ($n$ = 18 (29), with screening failures due to life expectancy < 3 months ($n$ = 2), increased risk of bleeding ($n$ = 11), patient not giving consent ($n$ = 4), and pregnancy ($n$ = 1)), and enrolled 45 in this prospective single-arm study of ultrasound-facilitated and catheter-directed low-dose fibrinolysis (Figure 2). The study patients were recruited from Schoen Hospital Neustadt in Holstein, Germany.

This work was approved by the institutional review board at Luebeck University with approval reference number 21–172, and written informed consent was obtained for every patient.

This study was designed and conducted by an experienced cardiologist at Schoen Hospital Neustadt in Holstein, Germany. Executive colleagues established the sample size and data analysis plan and participated in the statistical analysis in cooperation with the Institute of Medical Biometry and Statistics at Luebeck University in Germany.

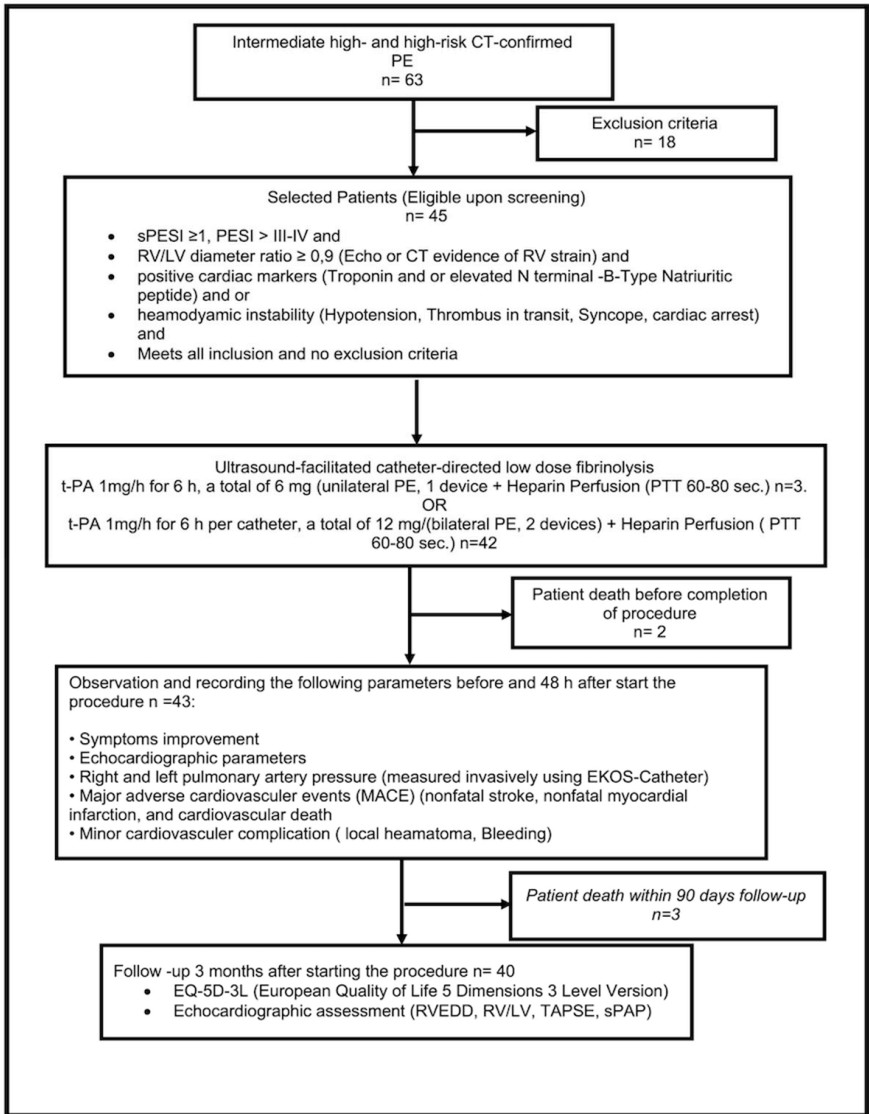

**Figure 2.** Study flow chart through each stage of the study. Overall, sixty-three patients were screened for eligibility. Forty-five patients were eligible for enrollment. Two patients died before completion of therapy and three patients died within 90 days.

### 2.2. Sample Size Calculation

The Ultima study [7] reported improvements in systolic (12.3 ± 10.0 mm Hg), diastolic (3.2 ± 7.8 mm Hg), and mean (5.7 ± 7.6 mm Hg) values. With a significance level of 5% and power of 80%, the smallest difference (diastolic) requires 39 analyzable patients "https://rpact.shinyapps.io/public (accessed on 6 August 2020)". Thus, least 40 should be included. The sample size was calculated for a one-sample *t*-test (one-sided): H0mu = 0; H1: effect = 3.2; standard deviation = 7.8; power: 80%.

### 2.3. Statistics

The results for continuous variables were compared with baseline using a paired *t* test or Wilcoxon signed rank test. For two-group comparisons, a two-sample *t* test or Wilcoxon rank sum test was used for continuous data, and the Fisher exact test was used for binary data. All reported *p* values were two-sided, and a *p* value < 0.05 was considered statistically significant. An intercept-only mixed model repeated-measures analysis was conducted, changing the RV/LV diameter ratio, and TAPSE and systolic pulmonary artery pressure were measured indirectly in the echocardiography as the dependent variable. The results

were adjusted for the logarithm of NT-Pro-BNP at baseline; *p* values were not adjusted for the multiplicity of comparisons between different points in time, as there were just two or three time points.

To compare all measurement times simultaneously, ANOVA or the Friedman Test with multiple post hoc comparisons was performed. To assess the statistical significance of single comparisons, *p*-values with the Bonferroni–Holm correction were used. All statistics were determined using jamovi version 1.6 statistical software retrieved from "https://www.jamovi.org (accessed on 26 May 2022)".

All statistical calculations were performed using the statistical software package SPSS (Version 28).

### 2.4. Study Population

Patients were eligible to participate if they had the following:

- An intermediate–high risk of PE (Simplified Pulmonary Embolism Severity Index (sPESI) $\geq 1$, RV strain, positive cardiac markers);
- High risk of PE (with systemic arterial hypotension, cardiogenic shock, or resuscitated);
- Intermediate–low risk of PE with deterioration of symptoms or hemodynamics (systemic arterial hypotension, cardiogenic shock, oxygen Saturation < 90%, heart rate > 100/min, and breathing frequency > 20/min).

Exclusion criteria were as follows:

- Ischemic stroke within three months;
- Active intracranial or intraspinal disease within 6 months;
- Major surgery within 7 days;
- Recent active bleeding from a major organ;
- Platelets < 50,000/mL;
- Pregnancy.

### 2.5. Anticoagulation

Anticoagulation was initiated with full-dose intravenous unfractionated heparin with a target-activated partial thromboplastin time (PTT) of 60 to 80 s. After completion of the procedure, all subjects received full anticoagulation. The drug, dose, frequency, and duration of anticoagulation were selected by the attending physician and according to the patient's comorbidity.

### 2.6. Ultrasound-Facilitated and Catheter-Directed Low-Dose Fibrinolysis

The EkoSonic Endovascular System (EKOS, Boston Scientific, Marlborough) comprises three components: the Intelligent Drug Delivery Catheter, a removable MicroSonic device, and a reusable Eko-Sonic control unit. The procedure was performed by an experienced operator from interventional cardiology.

Venous access was obtained via the common femoral route. After catheter placement and before the activation of the ultrasound and infusion of fibrinolytic therapy, baseline right and left pulmonary artery pressures were transduced through the catheters (Figure 3). Because the catheters were placed in or adjacent to the PE, the fibrinolytic agent was delivered directly to the thrombus. The fixed-dose regimen of tissue-plasminogen activator (t-PA) was 6 mg at 1 mg/h with saline coolant at 35 mL/h for unilateral PE (one drug delivery device was placed) and 12 mg at 1 mg/h with saline coolant at 35 mL/h for bilateral PEs (two drug delivery devices were placed). The fibrinolytic agent was infused under pressure using a perfusion pump, and a continuous infusion of t-PA was administered in each catheter for 6 h. No adjunctive interventional techniques to assist thrombus removal or dissolution were permitted. During the procedure, intravenous unfractionated heparin was continued according to the target PTT of 60 to 80 s.

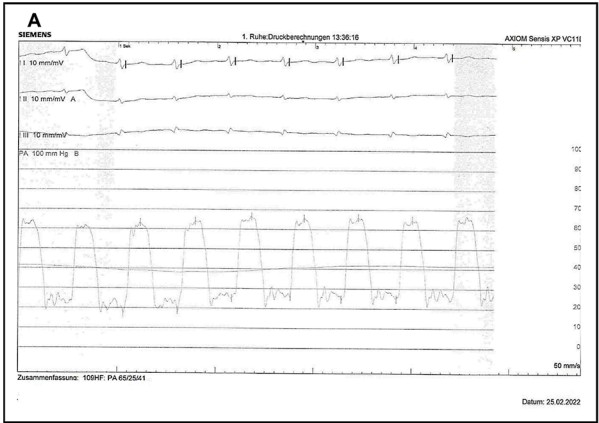
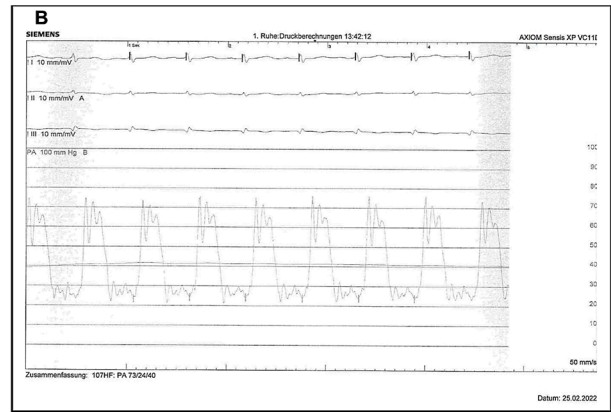

**Figure 3.** Right (**A**) and left (**B**) invasive measurement of pulmonary artery pressure before initiation of therapy.

*2.7. Follow-Up*

See Figure 4 and S2; the sheath was removed 4 h after termination of EKOS therapy, the access site was manually compressed for at least 10 min, and the compressing band was left for about 6 h. Full therapeutic anticoagulation was continued.

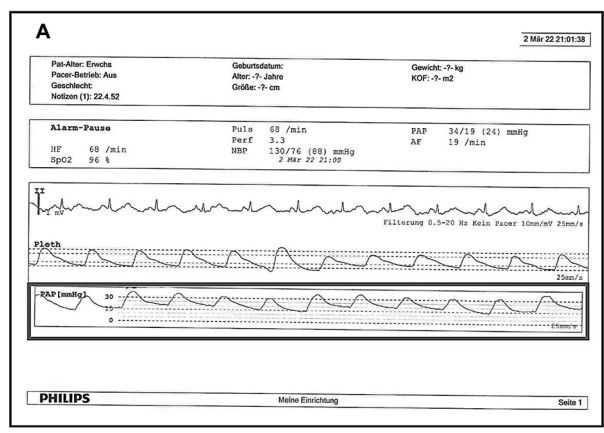
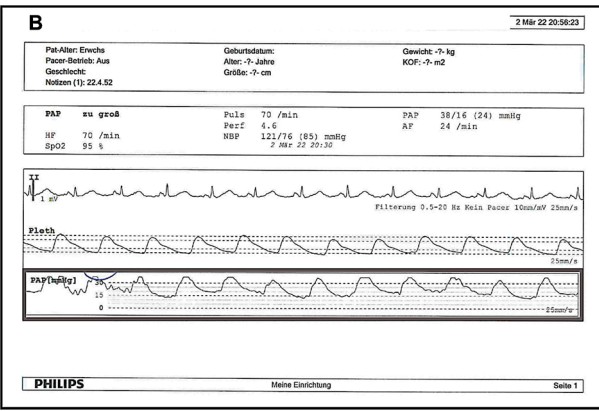

**Figure 4.** Right (**A**) and Left (**B**) invasive pulmonary artery pressure measurement 6 h after initiation of therapy at ICU.

Follow-up transthoracic echocardiography was performed within 48 h after initiation of the procedure to evaluate the changes in the RV/LV diameter, ratio, and RV function using tricuspid annular plane systolic excursion (TAPSE) and to estimate pulmonary artery systolic pressure noninvasively. Improvement in symptoms and hemodynamic assessment were recorded and analyzed by experienced cardiologists. Bleeding complications and safety outcomes were assessed up to 48 h after the procedure.

Patients were scheduled for a 90-day follow-up clinical visit; the repeated echocardiography and assessment of the outcomes are detailed below.

## 3. Results

*3.1. Outcomes*

1. The primary efficacy was the mean difference in the right and left pulmonary artery pressures measured invasively using the fibrinolysis delivery catheter at baseline to 6 h after initiation of therapy in the intensive care unit.

2.    A secondary efficacy outcome was the difference in RV/LV diameter ratio, right ventricular function, and pulmonary artery systolic (measured noninvasively) from baseline to 48 h after the initiation of the procedure and at 90 days.

3.    The primary safety outcome was death, as well as major and minor bleeding within 48 h of the initiation of the procedure. Bleeding events were classified by the GUSTO (Global Utilization of Streptokinase and Tissue Plasminogen Activator for Occluded Coronary Arteries) bleeding criteria [9]. Severe or life-threatening bleeding was defined as either intracranial hemorrhage or bleeding that caused hemodynamic compromise and required intervention. Moderate bleeding required a blood transfusion without resulting in hemodynamic compromise. Mild bleeding did not meet criteria for either severe/life-threatening or moderate bleeding. In addition, we assessed all-cause mortality at hospital discharge and through 90 days, the symptomatic recurrent PE up to 90 days after the initiation of the procedure, and procedural complications. Mortality was further classified as related to cancer, PE, stroke, or other causes.

4.    The secondary safety outcome evaluated quality of life assessment using the German version of the EQ-5D-3L questionnaire [10].

### 3.2. Baseline Demographics and Clinical Characteristics

The mean age was $66.7 \pm 14.7$ years (Table 1). The mean body mass index ($29 \pm 6.8 \text{ kg/m}^2$) was consistent with an obese patient population. Common risk factors for PE included obesity (73%), hypertension (41%), immobility within 90 days of PE diagnosis (17%), chronic lung disease (20%), presence of deep vein thrombosis (DVT) (48%), chronic renal disease (11%), previous history of stroke (11%), present history of active cancer (22%), and hyperlipidemia (13%).

**Table 1.** Baseline Demographics and Clinical Characteristics.

| | |
|---|---|
| Age | $66.8 \pm 14.7$ |
| Height in cm | $173.8 \pm 9.7$ |
| Weight in Kg | $88.5 \pm 21.3$ |
| Body mass index kg/m$^2$ | $29.0 \pm 6.8$ |
| Female | 23 (51.1) |
| Ethnicity/race | 45 (100) Caucasian |
| Comorbid conditions | |
| Hypertension | 18 (41) |
| Diabetes Mellitus | 2 (4.4) |
| Smoking | 3 (6.7) |
| Obesity | 32 (73) |
| Immobility | 7 (16) |
| Surgery | 2 (4.4) |
| Previous TVT | 2 (4.4) |
| Previous PE | 1 (2.2) |
| Atherosclerotic cardiovascular disease | 3 (6.7) |
| Family history of venous thromboembolism | 3 (6.7) |
| Stroke | 5 (11) |
| Active cancer | 10 (22) |
| Chronic pulmonary disease | 9 (20) |
| Renal insufficiency (GFR < 90 mL/min) | 5 (11.1) |
| Hyperlipidaemia | 6 (13.3) |
| Rheumatic disease | 4 (8.8) |
| Estrogen use | 2 (4.4) |
| Congenital Blood disease | 1 (2.2) |

Values are mean $\pm$ SD or *n* (%).

### 3.3. Baseline Characteristics of Pulmonary Embolism

All patients had symptomatic PEs (Table 2). The duration of PE symptoms was within 14 days. The diagnosis was established using contrast CT-Thorax with contrast. Intermediate- and high-risk PE and high-risk PE were observed in 40 (88) and 5 (11) patients, respectively.

**Table 2.** Characteristics of PE.

| Clinical presentation of PE | |
|---|---|
| Dyspnoea II | 1 (2.2) |
| Dyspnoea III | 24 (55) |
| Dyspnoea IV | 8 (18) |
| Chest pain | 1 (2.2) |
| Cardiopulmonary resuscitation | 3 (6.7) |
| Syncope | 3 (6.7) |
| Shock | 5 (11) |
| Palpitation | 7 (16) |
| Dizziness | 2 (5) |
| Cough | 1 (2.2) |
| Duration of symptoms/ days (maximum) | <14 days |
| **PE risk stratification** | |
| High risk | 5 (11) |
| Intermediate high risk | 40 (88) |
| Diagnosed TVT | 22 (48), right 11 (24), left 11 (24) |
| Oxygen saturation | 88.8 ± 7.7 |
| Breathing frequency | 21.2 ± 5 |
| Heart frequency before therapy | 97.3 ± 17.3 |
| Heart frequency after therapy | 81.2 ± 7.1 (mean difference 16) |
| **Blood pressure/ mmHg** | |
| Systole | 137.6 ± 25.2 |
| Diastole | 82.4 ± 16.6 |
| **Computer tomography-PE** | |
| **Unilateral** | |
| Right | 3 (6.7) |
| Left | 1 (2.2) |
| Bilateral | 41 (91.1) |
| **ECG** | |
| Sinus rhythm | 20 (44) |
| Sinus tachycardia | 16 (36) |
| Atrial flatter | 2 (4.4) |
| Atrial fibrillation | 7 (15.6) |
| Right bundle branch block | 2 (4.4) |
| **Laboratory Parameters** | |
| Troponin, ng/mL | 504 |
| $n$ terminal-B-Type Natriuretic peptide, pg/ml | 4451 |
| Lactate | 2.6 ± 2.1 |
| Hemoglobin in g/dl before | 12.9 ± 2.2 |
| Hemoglobin in g/dl after Therapy | 11.2 ± 2.1 (mean difference 1.72, (1.27–2.17)) |

Values are mean ± SD or $n$ (%).

### 3.4. Procedural Characteristics

The mean total dose of t-PA was 6 or 12 mg (Table 3). All (100%) 86 devices were successfully placed. Right femoral venous access was the most frequently used for device placement (91%). Ultrasound guidance was used in 7% of vascular access procedures for catheter placement. Bilateral devices were placed in 91% of patients. Bilateral catheters were placed through a double-access site in every patient to treat bilateral PE.

**Table 3.** Procedural characteristics.

| Successful Device Placement | 86 (100) |
|---|---|
| **No. of devices per patient** | |
| 1 | 4 (8.9) |
| 2 | 41 (91.1) |
| **Access site** | |
| Right femoral vein | 41 (91.1) |
| Left femoral vein | 4 (8.9) |
| Completed infusion of t-PA | 43 (95.5) |
| Duration of investigation/Minute | 9 (3–35) |
| Radiation dosage in cGY/cm$^2$ | 1405 (305–3916) |

Values are mean $\pm$ SD or $n$ (%).

Two patients did not have follow-up invasive pulmonary artery and echocardiography measurements performed within the 48 h window due to death. One patient with high-risk PE died at the initiation of therapy with a baseline hemoglobin level of 6 mg/dl and severe comorbidity and short time of pulseless electrical activity. A second patient experiencing high-risk PE died due to retroperitoneal bleeding and septic shock four hours after the initiation of therapy.

Three patients with colon cancer, septic shock, and recurrent ischemic stroke died within 90 days. We estimated that at 6 h after the initiation of therapy (Table 4) there would be at least a decrease in the right systolic, diastolic and mean pulmonary artery pressures, from 43, 19, and 28 mmHg at baseline to 27, 13, and 18 mmHg, with mean differences of 16, 6, and 9.8, respectively, and a decrease in left systolic, diastolic and mean pulmonary artery pressures from 44, 21, and 29 mmHg at baseline to 29, 14, and 20 mmHg, with mean differences of 15, 6, and 9.6 mmHg, respectively (Table 4A; Figures 5, S1 and S2).

**Table 4.** Primary pulmonary artery pressure efficacy outcome.

| | **Outcome** | | |
|---|---|---|---|
| **PA/mmHg** | **Baseline** | **6 h** | **Mean Difference** |
| Right systole | 43.38 $\pm$ 14.66 | 27.62 $\pm$ 10.31 | 15.76 $\pm$ 14 (11.26–20.26) |
| Right diastole | 19.45 $\pm$ 7.98 | 13.45 $\pm$ 7.27 | 6.0 $\pm$ 8.1 (3.2–8.80) |
| Right mean | 27.90 $\pm$ 9.05 | 18.10 $\pm$ 6.92 | 9.8 $\pm$ 9.9 (6.71–12.91) |
| Left systole | 44.89 $\pm$ 14.31 | 29.49 $\pm$ 12.03 | 15.40 $\pm$ 14.4 (10.45–20.35) |
| Left diastole | 21.44 $\pm$ 7.28 | 14.97 $\pm$ 7.35 | 6.47 $\pm$ 9.3 (3.22–9.72) |
| Left mean | 29.91 $\pm$ 8.53 | 20.32 $\pm$ 8.46 | 9.59 $\pm$ 9.4 (6.28–12.90) |

Values are mean $\pm$ SD, mmHg, median and quartiles. All $p$ values < 0.0001. Patients with complete pulmonary artery measurements $n$ = 43; PA = Pulmonary artery.

The mean RV/LV decreased from 1.2 at baseline to 0.85 at 48 h, and to 0.76 at 90-day follow-up, with a mean difference of 0.48 ($p$ < 0.001) (Table 5; Figure 6). The RV diameter decreased from 43 mm at baseline to 35 mm within 48 h, with a mean difference of 8.40, and to 32 at 90-day follow-up with a mean difference of 11.46, a recovery of RV systolic function using TAPSE from 17.9 mm at baseline to 21 within 48 h and to 22 mm at 90-day follow-up.

Procedural success was achieved in 100% of cases (Table 6). The mean improvement of symptoms was 79 % at 90 days follow-up questionnaire, with mean quality of life 97% using the EQ-5D-3L questionnaire (100 = the best health you can imagine and 0 the worst health you can imagine).

Four moderate bleeding events occurred with discharge after intervention without any residuals (Table 7). It is difficult to infer the bleeding cause as either due to low-dose fibrinolysis or due to anticoagulation therapy, as the bleeding events occurred at the second

day post-EKOS therapy in two patients and in the other two patients due to complications at the access site.

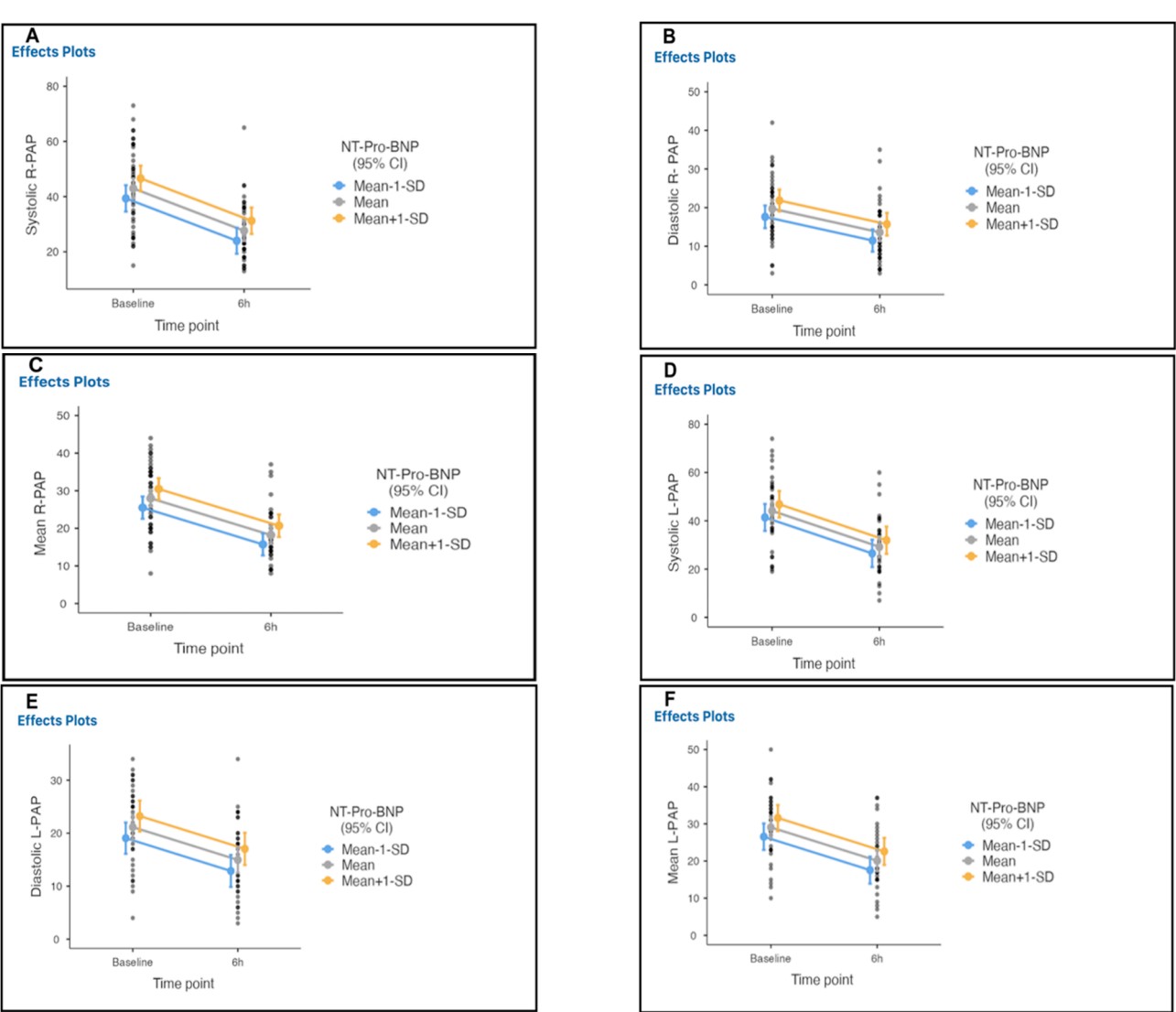

**Figure 5.** Invasive measurement of pulmonary artery pressure from baseline to 6 h after therapy initiation in all treated patients. Right systolic (**A**), diastolic (**B**), mean (**C**), left systolic (**D**), diastolic (**E**), and mean (**F**) pulmonary artery pressures significantly decreased from baseline to 6 h after therapy initiation in all treated patients.

**Table 5.** Primary echocardiography efficacy outcome.

| | | | Outcome | | |
|---|---|---|---|---|---|
| **Echocardiography** | **Baseline** | **48 h** | **Difference from the Baseline** | **90 Days** | **Difference from 48 h** |
| RVEDD, mm | 43.49 ± 4.44 | 35.1 ± 4.1 | 8.4 | 32 ± 3.3 | 3.1 |
| LVEDD, mm | 37.13 ± 7.14 | 42.38 ± 5.99 | −5 | 43.26 ± 5.67 | −6 |
| RV/LV ratio | 1.24 ± 0.28 | 0.85 ± 0.12 | 0.38 (0.27–0.50) | 0.76 ± 0.13 | 0.48 (0.37–0.59) |
| TAPSE, mm | 17.8 ± 5.2 | 21.4 ± 3.2 | −3.6 | 22.3 ± 2.9 | −0.9 |
| SPAP, mmHg | 49 ± 7.8 | 30.1 ± 7.2 | 19 | 26 ± 5.4 | 4 |

Values are mean ± SD. All *p* values < 0.0001. Patients with complete echocardiography measurements at 48 h *n* = 43, and at 90 days *n* = 40, SPAP = systolic pulmonary artery measurement; RV/LV = right ventricle-to-left ventricle.

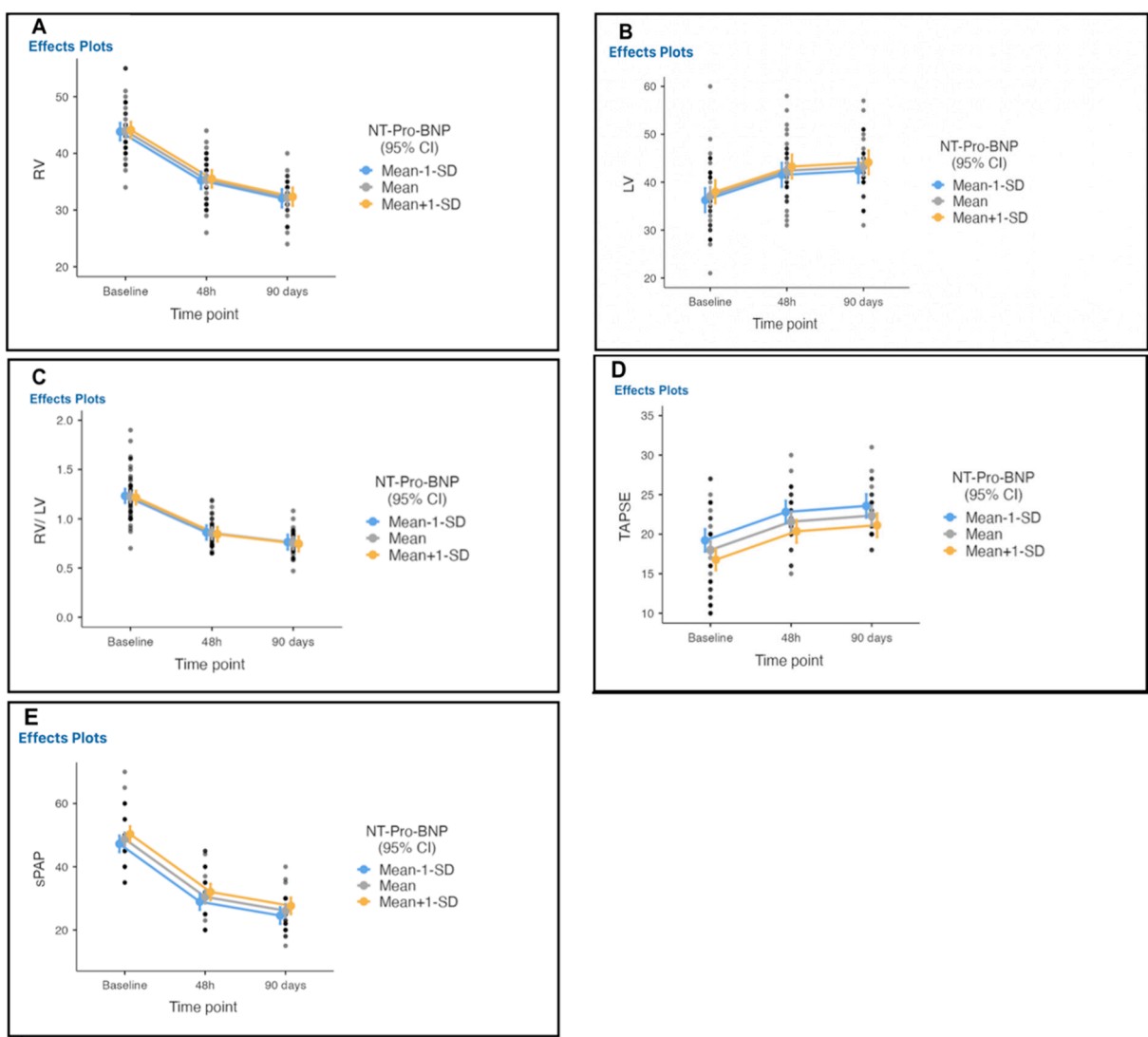

**Figure 6.** Echocardiographic parameters measured from baseline to 48 h and 90 days after therapy initiation. (**A**) RV diameter decreased, (**B**) LV diameter increase, (**C**) RV/LV ratio decrease, (**D**) improvement of TAPSE, and (**E**) noninvasively measured pulmonary artery pressure decreased significantly from baseline to 48 h and at 90 days follow-up after therapy initiation.

**Table 6.** Safety outcomes (*n* = 39 (%)).

| | |
|---|---|
| Length of Stay/Days | 6 (1–17) |
| In-hospital death, | 3 (6.7) |
| 90-day mortality, | 5 (11) |
| Serious and severe adverse events potentially related to device | 0 |
| Serious and severe adverse events potentially related to t-PA, | 1 (2.2) |
| Major bleeding within 90 days | |
| GUSTO moderate | 4 (8.8) |
| GUSTO severe | 1 (2.2) |
| Intracranial hemorrhage | 0 |
| EQ-5D-3L, mean ± SD | 0.97 ± 0.05 |
| Patient reported subjective clinical improvement; /%, mean ± SD | 78.71 ± 11.8 |
| No complication (*n*, %) | 34 (75.6) |
| 6MWT (*n*, %) | 42 (93.3) |

Values are mean ± SD, *n* (%), median and quartiles.

**Table 7.** Summary of GUSTO moderate major bleeds.

| Bleeding Event | Site of Bleed | Transfused Blood Products | Hg b (mg/dL) Baseline-24 h | Outcome and Complications |
|---|---|---|---|---|
| Access site hematoma | 4 (8.9) | | 11.6–11.3 | Recovered |
| | | | 12.8–10.1 | Recovered |
| | | | 18.7–15.2 | Recovered |
| | | | 17.6–14.1 | Recovered |
| Access site pseudoaneurysm | 3 (6.7) | | 14–11.4 | Thrombin injection Recovered |
| | | | 13.4–12.5 | Recovered |
| | | | 15–13 | Recovered |
| Access site bleeding | 1 (2.2) Retroperitoneal | 2-U PRBCs | 15.5–6.3 | Died of PE, comorbidity, and Shock |
| Mucosal bleeding | 1 (2.2), Unclear | 2-U PRBCs | 7.5–6.8 | Died of PE and comorbidity |
| | 1 (2.2) Rectal hematoma | | 12.1–10.8 | Rectal drainage, Recovered |
| Left Hemothorax | 1 (2.2) 2nd day after Lung resection, under Heparin | 1-U PRBCs | 11–8.7 | Thorax drainage, Recovered |
| Left Knee | 1 (2.2) Left Knee | | 14–11.4 | Knee Joint drainage, Recovered |
| Anemia | 2 (4.4) | 2-U PRBCs | | Recovered |

GUSTO–Global Utilization of Streptokinase and Tissue Plasminogen Activator for Occluded Coronary Arteries; Hgb—hemoglobin; PRBC—packed red blood cell.

## 4. Discussion

This study demonstrated that ultrasound-facilitated and catheter-directed fibrinolysis for the treatment of 45 patients with intermediate- to high-risk pulmonary embolism was associated with significant improvements in hemodynamics and few major bleeding events, as demonstrated in previous trials (Ultima and Seattle II). However, this protocol used lower doses and a shorter duration of t-PA treatment. The third arm of Optalyse PE-Trial used the same regime (6 mg/6 h/lung) for the treatment of 24 patients.

We focused on pulmonary artery hemodynamics as a primary precise surrogate marker for therapy effectiveness in addition to the RV/LV ratio that is used in previous trials (Ultima and Seattle II), as the latter has a bias in terms of interobserver variability.

As the primary increase in pulmonary artery pressure influences secondary changes in the RV/LV diameter ratio, we used both parameters as surrogate markers for primary endpoints to assess the effectiveness of ultrasound-facilitated and catheter-directed low-dose fibrinolysis.

This study evaluates the first-time use of fibrinolysis delivery catheters as a diagnostic as well as a therapeutic tool to assess the safety and effectiveness of this therapy.

This resulted in no episodes of intracranial hemorrhage. There was one GUSTO (Global Utilization of Streptokinase and Tissue Plasminogen Activator for Occluded Coronary Arteries)-defined major bleeding event (2.2) and four GUSTO-defined moderate bleeding events (8.9). We observed a significant decrease in pulmonary arterial pressure by more than 30% at the end of the procedure using a fibrinolysis delivery catheter, and a significant 30% decrease in the echocardiographically assessed RV/LV diameter ratio over 48 h.

The goal of this therapy is to restore reperfusion, decrease pulmonary artery pressure, recover RV function, and induce increases systemic perfusion, thereby improving symptoms and survival and preventing chronic thromboembolic pulmonary hypertension, which has its own comorbidities. In this study, we stress pulmonary artery pressure as a surrogate marker to predict and diagnosis this disease early. However, if CTEPH occurs, early detection is essential for optimal outcomes. The diagnostic criteria of CTEPH are as follows:

(1) Precapillary pulmonary hypertension (defined as a mean pulmonary arterial pressure (mPAP) $\geq$ 25 mmHg with a capillary pulmonary pressure $\leq$ 15 mmHg) assessed by right cardiac catheterization;

(2) At least one segmental perfusion defect evident on ventilation scan/perfusion;

(3) Signs of chronic thromboembolism on chest CT angiography or conventional pulmonary angiography (ring stenosis, chronic lesions, and/or complete occlusions) in patients receiving effective anticoagulation therapy for at least three months after a pulmonary embolism episode [11]. A recent detected soluble suppression of tumorigenesis 2 (sST2) concentration is higher in CTEPH patients treated with balloon pulmonary angioplasty (BPA) with complications in the postprocedural period. Thus, it could be helpful as an additional noninvasive marker for complications and risk stratification in patients undergoing BPA [12].

In the European ULTIMA (Ultrasound Accelerated Thrombolysis of Pulmonary Embolism) trial of 59 patients with submassive PE, ultrasound-facilitated and catheter-directed low-dose fibrinolysis plus anticoagulation used a higher dose of t-PA (20 mg), demonstrated a 0% incidence of major bleeding, improved RV function with a 30% difference in the RV/LV ratio, and reduced pulmonary artery pressure (using right heart catheter) from baseline to 24 h to a greater extent than anticoagulation alone [7]. In the Seattle II trial, with a population of 149 patients with submassive or massive PE, a slightly higher dose of t-PA (24 mg) was used, resulting in a 10% major bleeding rate, absence of intracranial hemorrhage, a 42% reduction in RV/LV ratios, and a decrease in pulmonary arterial pressure by 29% from baseline to 48 h post-procedure (Table 8) [13].

**Table 8.** Summary of main differences in EKOS trials and current study.

| Study | Ultima Trial (2014) | Seatlle II Trial (2015) | Optalysis Trial (2018) | Current Study (2022) |
|---|---|---|---|---|
| Nummer of Patient | $n = 59$ 30 in USAT 29 Heparin | $n = 149$ single arm | $n = 101$ Arm 1 $n = 28$ Arm 2 $n = 27$ Arm 3 $n = 28$ Arm 4 $n = 18$ | $n = 45$ single arm |
| Duration of therapy | $15 \pm 1$ h | 24 h (unilateral) 12 h (bilateral) | 2–6 h | 6 h |
| t-PA total dosis in unilateral PE | 10 mg | 24 mg | 4–12 mg | 6 mg |
| t-PA total dosis In bilateral PE | 20 mg | 24 mg | 8–24 mg | 12 mg |
| Safety: Major bleeding | No major bleeding | 10% major bleeding | 4% major bleeding | 2% major bleeding |
| Intracraniel heamorrahge | No | No | 2 Cases | No |
| Effectiveness | Pulmonary artery pressure (s/d/m) by right heart catheter RV/LV ratio No Echocardiography | Pulmonary artery systolic pressure by right heart catheter RV/LV ratio Control CTPA Echocardiography | No RV/LV ratio Control CTPA Echocardiography QoL | Pulmonary artery pressure (s/d/m) by drug delivery catheter RV/LV ratio No Echocardiography QoL |

The difference in the definitions of major bleeding in previous studies explains the variation in outcomes (0% in the ULTIMA study, 10% in the Seattle II study, and 2% in this study).

Our study used a lower dose of t-PA (6 mg/lung/6 h). This regime was used in the third arm of the OPTALYSE PE trial for 24 patients, demonstrating that treatment with ultrasound-facilitated catheter-directed low-dose fibrinolysis using a shorter delivery duration and lower dose of t-PA (4–12 mg of t-PA for 2.6 h) had similar efficacy and better safety outcomes compared to higher doses (one episode of ICH in the arm of 12 mg/lung/6 h and 4% overall risk of bleeding) [14].

In this study, as in the ULTIMA and Seattle studies, invasive hemodynamic measurements using fibrinolysis delivery catheter confirmed a significant reduction in pulmonary artery pressure and improvement of echocardiographic parameters after the completion of therapy. It is suggested that ultrasound-facilitated and catheter-directed fixed lower-dose fibrinolysis is similarly effective for reducing pulmonary artery pressure and improving echocardiographic parameters within 6 and 48 h, respectively, as compared to the higher-dose fibrinolysis used in the Ultima (20 mg t-PA) and Seattle (24 mg t-PA) studies.

In contrast to the ULTIMA and Seattle study protocols, the current study used EKOS fibrinolysis delivery catheter as a diagnostic tool to measure the pulmonary artery pressure invasively before and 6 h after the initiation of the procedure. The results matched the right heart catheter results in the EKOS and Heparin arm in the Ultima trial, thereby confirming the validity of this diagnostic tool to assess therapy effectiveness and to identify PE with risk of adverse outcomes.

The apparent reduction related complications of additional procedures, hospital residency, and economics gives this therapy additional diagnostic potential. Further, it stresses the importance of an interdisciplinary team in the management of PE.

On 21 May 2014, based on the data from Ultima, Seattle II, and PEITHO trials along with previous studies, the U.S. Food and Drug Administration approved the Eko-Sonic Endovascular System for the treatment of PE [6,15]. We thereby demonstrated that the early stabilization of hemodynamics within a short duration (6 h) with similar safety and effectiveness outcomes allowed early patient referral from the ICU to internal departments. We strongly recommend the routine measurement of invasive pulmonary artery pressure as an important surrogate marker in the evaluation of therapy effectiveness. However, the long-term benefits of early hemodynamic improvement after reperfusion treatment remain unclear.

The current ESC guidelines [4] recommend the use of systemic full-dose fibrinolysis in high-risk PE patients. However, its use even in the highest-risk patients has decreased over the past two decades due to fibrinolysis contraindication at the time of clinical presentation and the higher rate of intracranial hemorrhage as compared to anticoagulation therapy alone [15]. In an intermediate risk group, being the largest study of 30 to 50 mg dose systemic fibrinolysis, Tenecteplase reduced the risk of death or cardiovascular collapse by 56% in 1006 patients [16]. This benefit, however, was offset by a nearly five-fold increase in major bleeding risk and a ten-fold increased risk of hemorrhagic stroke. Meta-analyses of trials of systemic fibrinolysis for acute PE have demonstrated similar findings [17,18].

Our study expands the experience of ultrasound-facilitated and catheter-directed low-dose fibrinolysis in patients with intermediate–high and high-risk PE and demonstrates the potential of this technique, even in high-risk PE patients with moderate risk of bleeding. In the overall safe population, neither ICH nor the recurrency of PE was observed. Deaths were observed for two patients early after therapy (high-risk PE, comorbidity, and one major bleeding event) as well as three patients by 90 days (advanced-stage colon cancer, septic shock, and recurrent ischemic stroke).

This study demonstrated that a lower dose and short duration of t-PA treatment was associated with significant improvements in hemodynamics and few major bleeding events. Our study took into consideration the quality of life within 90 days using the German EQ-5D-3L questionnaire. The OPTALYSE-PE trial demonstrated an improvement in functional performance, 6MWT, and quality of life from the interval of 1 month to 1 year after initial therapy [14]. Finally, strategies to reduce major and moderate bleeding related to the procedure should be taken into consideration, for example, ultrasound-assisted puncture.

*Study Limitations and Strengths*

The major limitation of this study was the lack of a control group. Therefore, we cannot comment on the effectiveness and safety of fibrinolysis using a delivery catheter as a diagnostic tool to evaluate hemodynamics compared with anticoagulation alone or with other interventional modalities. However, the Ultima trial reported significant improve-

ments in hemodynamics using right heart catheterization, CT, and an echocardiographic parameter in the USCDT and Heparin group, unlike the anticoagulation one.

Another potential treatment is catheter-directed low-dose fibrinolysis therapy without the use of ultrasound to investigate and understand the impact of ultrasound facilitation, especially in chronic intermediate PE or in cancer patients, where the thrombus is more organized and fibrotic than in acute PE.

An important limitation was the lack of a right heart catheter in follow-up for estimation of pulmonary artery systolic pressure within 90 days, as was the lack of a randomized design to minimize bias.

The population was racially and ethnically homogenous as a solely Caucasian population, with no other groups present.

In summary, this protocol used a lower dose and shorter duration of t-PA in 45 PE patients, demonstrating similar safety and effectiveness outcomes. It focused on pulmonary artery hemodynamic as a primary precise surrogate marker for therapy effectiveness assessed by fibrinolysis delivery catheter. It confirmed the validity of this diagnostic tool and stressed the importance of an interdisciplinary team in the management of PE.

## 5. Conclusions

Ultrasound-facilitated catheter-directed low-dose fibrinolysis does not result in intracranial hemorrhage, reduces pulmonary artery pressure, decreases the RV/LV diameter ratio, improves the RV function, relieves symptoms in acute intermediate high-risk PE, and improves quality of life. In addition to these therapeutic benefits, an EKOS fibrinolysis delivery catheter can be used as a valid diagnostic tool to assess therapy effectiveness by measuring pulmonary artery hemodynamics. Lower doses and short duration of therapy could further reduce the complication with similar efficacy outcome, thus improve health, economics outcomes and shortens ICU stay.

## 6. Clinical Perspectives

### 6.1. Competency in Medical Knowledge

This study demonstrated that ultrasound-facilitated catheter-directed fibrinolysis for the treatment of 45 patients with intermediate- to high-risk pulmonary embolism was associated with significant improvements in hemodynamics and few major bleeding events, as demonstrated in previous trials (Ultima and Seattle II). However, this protocol used lower doses and a shorter duration of t-PA treatment, which shortens ICU stay to 6 h.

This study focuses on pulmonary artery hemodynamic measured by a fibrinolysis delivery catheter as a primary precise surrogate marker to ensure effectiveness, in addition to the RV/LV ratio that was used in previous trials (Ultima and Seattle II), which has a bias of interobserver variability.

This is a valid and precise diagnostic tool that can be used to assess therapy effectiveness and identify PE cases at risk of adverse outcomes such as chronic thromboembolic pulmonary hypertension (CTEPH), which may cause significant morbidities, as well as reduce the related complications of additional procedures, hospital residency, and economics. This gives this therapy potential as an additional diagnostic.

Our results stress the importance of an interdisciplinary team in management of PE, and the study evaluated the quality of life of the patients. This protocol shortens ICU stay to 6 h.

### 6.2. Translational Outlook 1

Although ultrasounds separate fibrin strands, enhancing the penetration of the thrombolytic agent into the emboli through low-dose fibrinolysis applied locally to reduce the bleeding complication, reduce the dose of administration, and at the same time achieve a reduction in the pulmonary artery pressure, it is paramount to assess the efficacy and safety outcomes of this therapy with other interventional modalities in order to better understand the optimal use of this therapy for acute PE.

Ultrasound-guided puncture should be assessed to identify whether this tool influences major and minor adverse outcomes in PE or not.

### 6.3. Translational Outlook 2

Pulmonary artery hemodynamic assessments using the EkoSonic Endovascular System fibrinolysis delivery catheter should be evaluated in a randomized trial investigating whether the treatment of high-risk PE favorably influences major adverse outcomes compared to standard therapy (anticoagulation alone). A HI-PEITHO study is underway [19].

**Supplementary Materials:** The following supporting information can be downloaded at: https://www.mdpi.com/article/10.3390/arm90060055/s1, Figure S1: Pulmonary artery pressure measurement before therapy initiation; Figure S2: Pulmonary artery pressure measurement 6h after therapy initiation at ICU.

**Author Contributions:** Conceptualization, A.E. and P.R.; methodology; A.E. and P.R.; software, A.E., M.E.; validation, P.R. and J.P.; formal analysis, A.E.; investigation, A.E., P.R., U.B. and M.K.; writing—original draft preparation, A.E. and J.P.; writing—review and editing, A.E., M.E. and P.R.; visualization, A.E. and M.E.; supervision, P.R.; project administration, A.E. All authors have read and agreed to the published version of the manuscript.

**Funding:** This research received no external funding.

**Institutional Review Board Statement:** Study patients were recruited at Schoen Hospital Neustadt in Holstein, Germany. This study was conducted in accordance with the Declaration of Helsinki and approved by the Institutional Review Board (or Ethics Committee) at Luebeck University (protocol code 21-172, 8.2020). All procedures performed in studies involving human participants were in accordance with the ethical standards of the institutional and/or national research committee and with the 1964 Helsinki Declaration and its later amendments or comparable ethical standards. Informed consent was obtained from all individual participants involved in the study.

**Informed Consent Statement:** Informed consent was obtained from all subjects involved in the study. Written informed consent was obtained from the patients to publish this paper.

**Data Availability Statement:** Study patients were recruited at Schoen Hospital Neustadt in Holstein, Germany and written informed consent was obtained for every patient. The study was designed and executed by an experienced cardiologist at Schoen Hospital Neustadt in Holstein in Germany; executive colleagues established the sample size and data analysis plan and participated in the statistical analysis in cooperation with the Institute of Medical Biometry and Statistics at Luebeck University in Germany. All data were archived safely in a safe dataset during this study. All data are available on request taking into consideration the polices of patient privacy.

**Acknowledgments:** This work was supported by my wife Aline Wenner and my son Yousef. A big thanks is to Reinhard Vonthein (Luebeck University; Institute of Medical Biometrics and Statistics) for his efforts and patience during the statistical analysis.

**Conflicts of Interest:** The authors declare no conflict of interest. The authors whose names are listed in this article certify that they have no affiliations with or involvement in any organization or entity with a financial or nonfinancial interest in the subject matter or materials discussed in this manuscript.

## Abbreviations

| | |
|---|---|
| RVEDD | Right ventricle end diastolic diameter |
| LVEDD | Left ventricle end diastolic diameter |
| SPAP | Systolic pulmonary artery pressure |
| TAPSE | Tricuspid annular plane systolic excursion |
| ECG | An electrocardiogram |
| AF | Atrial fibrillation |
| RBBB | Right bundle branch block |
| CGY/cm$^2$ | The absorbed dose multiplied by the area irradiated, expressed in gray-centimeters squared |
| EQ-5D-3L | European quality–five dimension–three level |
| USCDT | Ultrasound catheter-directed therapy |

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
