# Peer review of "Using a Fibrinolysis Delivery Catheter in Pulmonary Embolism Treatment for Measurement of Pulmonary Artery Hemodynamics"

_arm, doi:10.3390/arm90060055_

Round 1
Reviewer 1 Report
Abstract: the content is clear and concise. However, I find the presence of both an "Abstract" and a "Highlights" section redundant. The central topic is expressed in different tones, risking confusing the reader. I would opt for a single solution, which encompasses both, without becoming verbose.
Introduction: this section provides an overview of methods of treating PE of thrombotic origin, with relative references to the guidelines currently available, as well as the objective of the study. I would add some more descriptive details regarding PE and its diagnostics. In the same way I would add some information relating to the pulmonary arterial pressure value, for example in which pathologies it can be significant and how it is usually measured. In order to discuss the previously described points, important and recent references are needed to be added, such as:
a- Giant Pulmonary Artery Thrombotic Material, Due to Chronic Thromboembolic Pulmonary Hypertension, Mimics Pulmonary Artery Sarcoma. Medicina 2021, 57, 992. https://doi.org/10.3390/medicina57090992
b-Chronic Thromboembolic Pulmonary Hypertension: An Update. Diagnostics (Basel). 2022;12(2):235. Published 2022 Jan 19. doi:10.3390/diagnostics12020235
c-Soluble ST2 as a Biomarker for Early Complications in Patients with Chronic Thromboembolic Pulmonary Hypertension Treated with Balloon Pulmonary Angioplasty. Diagnostics 2021, 11, 133. https://doi.org/10.3390/diagnostics11010133
Methods: the criteria for the inclusion and exclusion of patients and the methods of data collection are reported. The steps of anticoagulation and fibrinolysis performed are also described in detail. As for the "Follow-up" section, I believe the "Outcomes" paragraph should be placed among the results, considering that it deals with the effectiveness of the procedure performed.
Results: with the aid of graphs and tables, data relating to the patients examined, their initial characteristics and what was reported during the procedures performed are reported in a complete and exhaustive way.
Discussion: a comparison between this study and the previous ones carried out on the same topic is presented in a precise manner, with the necessary data. A scheme that summarizes the content itself could be useful, distinguishing similarities and differences between the studies, with relative advantages and disadvantages, for a faster and clearer understanding of the text. I appreciated the presence of a chapter on the limits of the study, which always provides new insights into the future for the subject in question.
Conclusion: this section takes up what was reported in the initial “Highlights” chapter, as well as the “Clinical perspectives” section. I believe it is appropriate to summarize the 3 in a single text that contains all the necessary information.
Finally, I think a reflection on the title is useful: if the primary objective of the study was the measurement of PAP, I believe no changes are necessary; if instead this is only one of the results obtained, as reported in lines 297-299, perhaps a title with a more generic intention would be worth considering.
Author Response
7.11.2022: Point-by-point response to the reviewer’s comments
Manuscript ID: arm-1969448
Type of manuscript: Article
Title: Using fibrinolysis delivery catheter in pulmonary embolism treatment
for measurement of pulmonary artery pressure
Authors: Abdelrahman Elhakim *, Peter Radke, Johannes Patzelt,
Ulrich, Boehmer, Martin Knauth, Mohamed Elhakim
Received: 7 October 2022
E-mails: aelhakim@schoen-klinik.de
Dear Editors,
thank you again for your peer review, your patience and your suggestions to improve this manuscript. I tried as possible to took into considerations all your peer comments and I am open for any other suggestions or recommendations to improve this manuscript further.
My point-by-point response to the reviewer’s comments are:
- Response to Editor`s office Comments
(I) Please revise your manuscript according to the referees’ comments and upload the revised file within 5 days.
Response 1: I revised my manuscript according to the referees’ comments. I took into considerations all your peer comments.
(II) Please use the version of your manuscript found at the above link for your revisions.
Response 2: I downloaded only the revised manuscript from the given link and worked on it.
(III) Please check that all references are relevant to the contents of the manuscript.
Response 3: I checked all references and found that all references are relevant to my manuscript. If you find any irrelevant references, please inform me to delete it/them.
(IV) Any revisions made to the manuscript should be marked up using the “Track Changes” function if you are using MS Word/LaTeX, such that changes can be easily viewed by the editors and reviewers.
Response 4: I used the “Track Changes” function during my revision to be easly viewed by editors and reviewers.
(V) Please provide a short cover letter detailing your changes for the editors’ and referees’ approval.
Response 5: I prepared this point-by-point response letter for the reviewers and editors.
(VI) We have found an increased similarity to already published materials:
https://www.sciencedirect.com/science/article/pii/S1936879815008353?via%3Dihub
https://www.sciencedirect.com/science/article/pii/S1936879815009012?via%3Dihub
thus we would like to draw your attention to the sections: outcomes, statistical analysis, results description
Response 6: I reformulated as possible the outcomes, statistical analysis and results description sections. If you still find an increased similarity to already published materials please let me know exactly which paragraphs have similarity and I will reformulate or delete them if they are a non necessary parts.
If one of the referees has suggested that your manuscript should undergo extensive English revisions, please address this issue during revision. We propose that you use one of the editing services listed at https://www.mdpi.com/authors/english or have your manuscript checked by a native English-speaking colleague.
Response 7: This manuscript has been addressed for an extensive english revision and was revised by MDPI’s english editor service. The cerfificate of editing is available on request.
- Response to reviewer 1 Comments
Abstract: the content is clear and concise. However, I find the presence of both an "Abstract" and a "Highlights" section redundant. The central topic is expressed in different tones, risking confusing the reader. I would opt for a single solution, which encompasses both, without becoming verbose.
Response 1: At the end of the part “Abstract”, I encompassed and summarized conclusion and highlights in one sentence.
Introduction: this section provides an overview of methods of treating PE of thrombotic origin, with relative references to the guidelines currently available, as well as the objective of the study. I would add some more descriptive details regarding PE and its diagnostics. In the same way I would add some information relating to the pulmonary arterial pressure value, for example in which pathologies it can be significant and how it is usually measured. In order to discuss the previously described points, important and recent references are needed to be added, such as:
a- Giant Pulmonary Artery Thrombotic Material, Due to Chronic Thromboembolic Pulmonary Hypertension, Mimics Pulmonary Artery Sarcoma. Medicina 2021, 57, 992. https://doi.org/10.3390/medicina57090992
b-Chronic Thromboembolic Pulmonary Hypertension: An Update. Diagnostics (Basel). 2022;12(2):235. Published 2022 Jan 19. doi:10.3390/diagnostics12020235
c-Soluble ST2 as a Biomarker for Early Complications in Patients with Chronic Thromboembolic Pulmonary Hypertension Treated with Balloon Pulmonary Angioplasty. Diagnostics 2021, 11, 133. https://doi.org/10.3390/diagnostics11010133.
Response 2: I added a more descriptive details regarding PE and its diagnostics in the introduction. In the same way I added some informations related to the pulmonary arterial pressure value, for example in which pathologies can be significant (CTEPH) and how it is usually measured. All the above suggested recent references were added to the introduction and discussion parts.
Methods: the criteria for the inclusion and exclusion of patients and the methods of data collection are reported. The steps of anticoagulation and fibrinolysis performed are also described in detail. As for the "Follow-up" section, I believe the "Outcomes" paragraph should be placed among the results, considering that it deals with the effectiveness of the procedure performed.
Response 3: "Outcomes" paragraph placed among the results, considering that it deals with the effectiveness of the procedure performed.
Results: with the aid of graphs and tables, data relating to the patients examined, their initial characteristics and what was reported during the procedures performed are reported in a complete and exhaustive way.
Response 4: thank you for your support.
Discussion: a comparison between this study and the previous ones carried out on the same topic is presented in a precise manner, with the necessary data. A scheme that summarizes the content itself could be useful, distinguishing similarities and differences between the studies, with relative advantages and disadvantages, for a faster and clearer understanding of the text. I appreciated the presence of a chapter on the limits of the study, which always provides new insights into the future for the subject in question.
Response 5: Table 7 added and summarized the content, distinguishing similarities and differences between the studies, with relative advantages and disadvantages, for a faster and clearer understanding of the text as requested.
Conclusion: this section takes up what was reported in the initial “Highlights” chapter, as well as the “Clinical perspectives” section. I believe it is appropriate to summarize the 3 in a single text that contains all the necessary information.
Response 6: conclusion is summarized in a single text that contains all the necessary informations.
Finally, I think a reflection on the title is useful: if the primary objective of the study was the measurement of PAP, I believe no changes are necessary; if instead this is only one of the results obtained, as reported in lines 297-299, perhaps a title with a more generic intention would be worth considering.
Response 7: I modified the titel into ” Using fibrinolysis delivery catheter in pulmonary embolism treatment for measurement of pulmonary artery hemodynamics” and replaced pressures by hemodynamics as a generic one.
Both the referees has suggested that your manuscript should undergo
extensive English revisions, please address this issue during revision. We
propose that you use one of the editing services listed at
https://www.mdpi.com/authors/english or have your manuscript checked by a
native English-speaking colleague.
Response 8: This manuscript has an extensive English revision and was revised by MDPI’s english editor service.
(C) Response to reviewer 2 Comments
Both the referees has suggested that your manuscript should undergo
extensive English revisions, please address this issue during revision. We
propose that you use one of the editing services listed at
https://www.mdpi.com/authors/english or have your manuscript checked by a
native English-speaking colleague.
Response 1: This manuscript has an extensive English revisions and was revised by MDPI’s english editor service.
Thank you in advance for your patience and support. Please inform me, if I should do any further improvements to reach your acceptance level for publication of this manuscript.
Sincerely,
Abdelrahman Elhakim
Cardiology department,
Schoen hospital neustadt in holstein,
Am Kiebitzberg 10, 23730, Germany,
TEL: 00491638542698
E-Mail: ayelhakim1985@yahoo.com
aelhakim@schoen-klinik.de

Reviewer 2 Report
- The article focuses on ultrasound thrombosis in patient with high risk PE using a lower dose of fibrinolytic agent showing good treatment hemodynamic response and no bleeding complications
- The sample is small however metrology and power were calculated with previous reports and finding statistical significance of the results
- Hypothesis what proved, I found no methodological inaccuracies,
- Figures and text available are important for the results and discussion.
- The manuscript gives new information for treatment of a known disease and it focuses is well established and confirmed
- There are some spelling errors but no significant that should be address
Author Response

(The authors gave the same response as above.)

Round 2
Reviewer 1 Report
Authors applied all suggested changes in my review report.
Reviewer 2 Report
All previous comments are address